# Effect of Training Load on Post-Exercise Cardiac Troponin T Elevations in Young Soccer Players

**DOI:** 10.3390/ijerph16234853

**Published:** 2019-12-02

**Authors:** Rafel Cirer-Sastre, Alejandro Legaz-Arrese, Francisco Corbi, Isaac López-Laval, Jose Puente-Lanzarote, Vicenç Hernández-González, Joaquín Reverter-Masià

**Affiliations:** 1National Institute of Physical Education of Catalonia (INEFC), University of Lleida (UdL), Partida la Caparrella s/n, E-25192 Lleida, Spain; fcorbi@inefc.es; 2Research Group Human Movement (RGHM), Universitat de Lleida (UdL), Plaça de Víctor Siurana, 25003 Lleida, Spain; vicens_h_g@didesp.udl.cat (V.H.-G.); reverter@didesp.udl.cat (J.R.-M.); 3Section of Physical Education and Sports, University of Zaragoza, Calle de Pedro Cerbuna, 50009 Zaragoza, Spain; alegaz@unizar.es (A.L.-A.); isaac@unizar.es (I.L.-L.); 4Lozano Blesa University Hospital, Avda. San Juan Bosco, 50009 Zaragoza, Spain; jjpuentel@gmail.com; 5Section of Physical Education, Universitat de Lleida (UdL), Plaça de Víctor Siurana, 25003 Lleida, Spain

**Keywords:** cardiac biomarkers, exercise physiology, maturation

## Abstract

Training load (TL) metrics are usually assessed to estimate the individual, physiological and psychological, acute, and adaptive responses to training. Cardiac troponins (cTn) reflect myocardial damage and are routinely analyzed for the clinical diagnosis of myocardial injury. The association between TL and post-exercise cTn elevations is scarcely investigated in young athletes, especially after playing common team sports such as soccer. The objective of this study was to assess the relationship between TL measurements during a small-sided soccer game and the subsequent increase in cTn in young players. Twenty male soccer players (age 11.9 ± 2 years, height 151 ± 13 cm, weight 43 ± 13 kg) were monitored during a 5 × 5 small-sided game and had blood samples drawn before, immediately after, and 3 h after exercise for a posterior analysis of high-sensitivity cardiac troponin T (hs-cTnT). Internal, external, and mixed metrics of TL were obtained from the rating of perceived exertion (RPE), heart rate (HR), and GPS player tracking. The results show that the concentration of hs-cTnT peaked at 3 h post-exercise in all participants. The magnitude of hs-cTnT elevation was mainly explained by the exercise duration in the maximal heart rate zone (Maximum Probability of Effect (MPE) = 92.5%), time in the high-speed zone (MPE = 90.4 %), and distance in the high-speed zone (MPE = 90.45%). Our results support the idea that common metrics of TL in soccer, easily obtained using player tracking systems, are strongly associated with the release of hs-cTnT in children and adolescents.

## 1. Introduction

Soccer is the most popular sport around the world and involves more than 270 million people, including players, referees, and coaches [1]. Monitoring training load (TL) has become essential for prescribing and quantifying training stimuli. Strength and conditioning coaches, researchers, and sports analysts establish and routinely supervise players’ TL to prevent injuries, prescribe training, and improve performance [2,3,4]. To this end, TL has been broadly summarized under two main constructs, namely external (eTL) and internal (iTL) training loads [3]. The former refers to externally measurable aspects of the work performed by a player during training, such as repetitions, distance, duration, or speed. The latter, on the other hand, refers to measures of physiological and psychological responses of the player during training, such as oxygen consumption, heart rate (HR), or the rating of perceived exertion (RPE) [2,3]. Internal TL is conceptually restricted to measures that can be used to prescribe training and then monitored during exercise. However, there are other measures taken a posteriori, such as the recovery heart rate, post-exercise heart rate variability, or post-exercise lactate concentration, which may also reflect a player’s response to the training session. These measures might be effective as surrogate metrics of iTL [3]. In addition, previous authors have proposed comprehensive metrics that combine external and internal measurements of TL, such as session RPE or Edwards’ TL [5,6]. These might be applicable as mixed metrics of TL.

Metrics of TL are diverse in terms of measurement error, practical convenience, required equipment, or the time needed to analyze data [2,3]. Such heterogeneity may complicate the selection (e.g., by coaches and analysts) of a set of metrics that meets each team’s needs. Coaches commonly prescribe training using external measures of TL to elicit internal responses in the player. Therefore, both constructs remain strongly associated since eTL appears to be the main determinant of iTL [3]. Although iTL correlates with eTL, the responses (iTL) of different players to a common exercise (eTL) might differ depending on their training status, nutrition, health, psychological status, or genetics [3,7]. This variability in the relationship between external and internal TL among different players might be considered to be a player’s responsiveness to a training stimulus [8].

Cardiac troponins (cTn) are heart contractile proteins, and their increased blood concentration reflects myocardial damage [9]. Moreover, when exceeding the upper reference limit (URL) determined for a healthy reference population, they are the preferred biochemical indicator for the diagnosis of myocardial infarction (MI) and acute myocardial infarction (AMI) [10]. Numerous studies have reported that healthy populations of all ages have an elevated blood concentration of cTn in the hours after intense exercise [11,12]. Thus, it has been suggested that exercise-induced releases of cTn might be related to a physiological rather than a pathological response to exercise [13,14]. Exercise-induced release of high-sensitivity cTn T (hs-cTn) has been associated with the intensity and volume of exercise [13]. However, previous analyses of exercise-induced hs-cTn have not accounted for most of the TL metrics that are routinely used to assess sports performance [4,15]. In this regard, it is worth noting that intensity and volume do not necessarily imply internal and external TL, respectively. Exercise intensity, for instance, might be reported in terms of relative heart rate (iTL) or relative running speed (eTL).

If cTn increases are related to a physiological response to exercise, then cardiac stress estimated from its release after exercise might be a suitable surrogate metric of iTL, together with recovery heart rate (HR), post-exercise heart rate variability (HRV), or post-exercise lactate. Immunoassays for cTn are becoming increasingly more affordable, less invasive, and quicker. Therefore, high-sensitivity cTn measurements might be applicable to sports performance as an objective marker of athletes’ cardiac response to exercise, as well as their tolerance to selected TLs. Furthermore, incorporating TL metrics into the investigation of post-exercise cTn kinetics could be used in clinical research for a better understanding of the phenomenon itself, as well as the potential clinical implications. For these reasons, the main purpose of this investigation was to assess the association between exercise-induced increases in hs-cTn and a set of TL metrics. Since hs-cTnT is a specific marker from myocardial tissue, our main hypothesis was that its exercise-induced release is explained fairly well by internal TL metrics that are based on heart rate activity during the exposure. Furthermore, since internal and external TL metrics are linearly associated, our secondary hypothesis was that hs-cTnT is also explained by the metrics related to external TL.

## 2. Experimental Section

### 2.1. Participants

The subjects in this study were a convenience sample of 20 participants, whose demographic characteristics are reported in Table 1. Subjects were recruited through an open invitation to all players participating in training at a soccer technical training campus (Campus de Futbol Formativo Pichi Alonso, Peñíscola, Spain). Parents were invited to fill in a digital form that included the Spanish version of the revised Physical Activity Readiness Questionnaire (PAR-Q) [16], as well as questions related to participants’ medical and training histories. Answers were used to screen volunteers who met the following criteria: favorable PAR-Q, male, aged under 19 years, at least 3 years of experience in competitive soccer, and a weekly training volume of at least 3 days/week. Parents were informed of the procedures involved and gave their prior consent. The procedures of this study were approved by the Ethical Committee of Clinical Research of Sports Administration of Catalonia (02/2018/CEICGC) and met the principles of the latest revision of the Declaration of Helsinki [17].

### 2.2. Design

This was an observational study with a repeated measures design.

### 2.3. Instruments

Body weight was measured with a medical scale (SECA 711, Hamburg, Germany), and height was measured with a wall stadiometer (Año-Sayol, Barcelona, Spain). Participants were equipped with a WIMU Pro™ (RealTrack Systems, Almería, Spain) GPS tracker (sampling rate = 10 Hz), which was synced with a Garmin™ heart rate band (Garmin, Ltd., Olathe, Kansas, US). This setup has been validated for both heart rate (*R*^2^ = 0.96) and player geospatial tracking (*ICC* = 0.98) in previous studies [18,19]. For high-sensitivity cTnT (hs-cTnT), blood samples were analyzed using a Troponin T hs STAT immunoassay in a Cobas E 601 analyzer (Roche Diagnostics, Penzberg, Germany). The detection range of this assay is 3–10,000 ng/L, and the intra-assay coefficient of variation at a mean hs-cTnT of 13.5 ng/L is <10%. The upper reference limit (URL) for hs-cTnT, defined as the 99th percentile of healthy participants, is 13.5 ng/L [20]. Concentrations below the limit of detection of 3 ng/L were set to 1.5 ng/L for statistical analyses. The rating of perceived exertion (RPE) was obtained immediately after exercise using the CR100 scale, and all participants were already familiar with RPE reporting [21].

### 2.4. Variables

The response variable was the hs-cTnT concentration before (Pre hs-cTnT), immediately after (Post0h hs-cTnT), and 3 h after exercise (Post3h hs-cTnT). Sampling times were chosen in accordance with previous research, which suggested that the peak hs-cTnT value occurs between 2 and 5 h after exercise [11,12]. Blood samples were drawn from an antecubital vein by a nurse and quickly centrifuged. Then, serum and plasma were drawn off and stored at −80 °C for further analysis. Participants were asked to refrain from exercise for 48 h before the intervention, as well as after exercise until the third blood sample was taken.

We grouped predictors for hs-cTnT changes over time into four categories: athlete profile, internal TL, external TL, and mixed metrics of TL. Each athlete profile comprised age (years), Tanner stage (II, III, or IV), training experience (years), and Fox’s maximum heart rate (bpm) [22]. Internal TL metrics were the average and peak HR, in both absolute (bpm) and relative (% HR max) terms, and the RPE (Arbitrary Units, AU) [21]. The maximum heart rates for relative intensities were calculated using Tanaka’s formula of 208 − (0.7 × Age). The external metrics of TL were the total distance covered during exercise (m), the relative distance covered per minute of exercise (m·min^−1^), the average speed during exercise (km·h^−1^), the maximum speed during exercise (km·h^−1^), the distance covered in the high-intensity speed zone (m at >18 km·h^−1^), and the time spent at a high-intensity speed (min at >18 km·h^−1^) [15]. Finally, two mixed metrics of TL were calculated, namely, the session RPE (AU) [5] and Edwards’ training load (AU) [6].

### 2.5. Experimental Setup and Protocol

Measurements were taken at the participants’ training facilities on two separate occasions. On the first visit, anthropometric measurements were taken, and a single experienced pediatrician classified participants according to their maturational stage [23]. After these measurements, participants were equipped with the tracking system and performed a familiarization session consisting of the standardized “11 + Kids” warm-up [24] and a simulation of the small-sided game to be performed on the second day.

On the second day, participants attended our facilities in two groups of 10, in a randomized order. On arrival, we extracted baseline blood samples (30 ± 5 min prior to exercise), and participants were equipped with the tracking system and then warmed up for 15 min by performing the “11 + Kids” exercises. Then, the cohort was randomly assigned to two teams of five to play a small-sided game of 5 × 5. Teams were consistent across bouts. The exercise consisted of 16 min of effort (four efforts of 4 min) alternated with 9 min of passive rest (three rest intervals of 3 min) [25]. The game was played on a pitch of 20 m × 30 m and consisted of passing the end lines with the ball. The offside rule was adopted. Researchers and coaches were placed on all four lines, and when a team scored or exceeded the demarcation limits, a new ball was quickly delivered to the other team [25]. Participants were asked to play at maximal intensity, and strong and standardized verbal encouragement was given during the match. Immediately after exercise, participants reported their RPE, and the second blood sample was taken (5 ± 5 min after exercise). Participants stayed in a resting room and avoided physical exercise until 180 ± 5 min after exercise when the third blood sample was taken. Players were allowed to drink water ad libitum. The timing of procedures is outlined in Figure 1.

### 2.6. Statistical analysis

Tracking data were downloaded onto a computer and extracted to a spreadsheet using S Pro ™ software (RealTrack Systems, Almería, Spain). All statistical analyses were then performed using R version 3.5.1 (R Foundation for Statistical Computing, Vienna, Austria). Bayesian models were created in the Stan computational framework [26] accessed with the brms package [27]. Concentrations of hs-cTnT were log-transformed prior to model fitting to achieve a residual normal distribution. Descriptive statistics are presented as the mean ± standard deviation or median (interquartile range). Statistical analyses were performed by fitting Bayesian linear mixed-effects models using Markov Chain Monte Carlo. Since variables were registered on very different scales, data were standardized prior to model fitting to facilitate model convergence and their posterior comparison. After verifying differences in hs-cTnT between baseline and peak values (3 h post), we proceeded by fitting separate models for each variable. The response variable was the hs-cTnT concentration (log-transformed). The reference values for the intercept (*β*_0_) were set to time = baseline and each variable at Z = 0. The fixed effects were time = Post 3 h (*β*_1_), each variable separately (*β*_2_), and their interaction (*β*_3_). A random effect for each participant was always included to account for individual variation. The goodness of fit of the models was assessed by the proportion of variance explained (*R*^2^). All model parameters (*β*_i_) are reported as the median (90% credibility interval), and the likelihood of *β* ≠ 0 is expressed as the Maximum Probability of the Effect (MPE). Effects were considered highly likely when the MPE was above 90%.

## 3. Results

Descriptive statistics of training load metrics are summarized in Table 2. Troponin concentrations were under the limit of detection (LoD) in 17 (85%), 15 (75%), and 3 (15%) participants before, immediately after, and 3 h after exercise, respectively. Four participants (20%) exceeded the URL for AMI 3 h after exercise.

The results of Bayesian models are presented in Appendix A. There were differences in hs-cTnT over time R^2^ = 64.4% (90% CI 52.5–74.7%). Immediate changes in hs-cTnT after exercise were shown to be unlikely (MPE = 84.4%). However, there was a high likelihood of an increase in hs-cTnT between baseline and 3 h after exercise (MPE = 100%). Subsequent models confirmed a highly likely effect of time (*β*_1_), regardless of the analyzed covariate (MPE = 100%). Moreover, any covariate by itself (*β*_2_) appeared to explain the hs-cTnT concentration. Figure 2 shows the interaction effect (*β*_3_) of each explanatory variable, as described below.

For the athlete profile, the increase in hs-cTnT could be explained by adding an interaction between time and age (R^2^ = 68.7%, MPE = 99.1%), Tanner’s maturational stage (R^2^ = 63.8%, MPE = 91.2%), years of training experience (R^2^ = 72.9%, MPE = 99.7%), and HR max (R^2^ = 69.1%, MPE = 98.8%). In contrast, an hs-cTnT increase was unlikely to be explained by participants’ BMI (R^2^ = 59.8%, MPE = 56%). Only two metrics of internal TL could explain the increase in hs-cTnT over time with a high likelihood, namely time spent in the high-intensity HR zone (R^2^ = 60.7%, MPE = 92.7%) and RPE (R^2^ = 66%, MPE = 96.1%). On the contrary, neither the average (R^2^ = 57.6%, MPE = 64.5%) and peak HR (R^2^ = 56.6%, MPE = 64.5%) nor the average (R^2^ = 58.9%, MPE = 78.4%) and peak relative HR (R^2^ = 57%, MPE = 55.3%) appeared to interact with the increase in post-exercise hs-cTnT. All external metrics of TL could explain the increase in hs-cTnT with a high likelihood: absolute (R^2^ = 70.1%, MPE = 99.4%) and relative distance (R^2^ = 69.8%, MPE = 99.5%); average (R^2^ = 71.7%, MPE = 99.7%) and peak speed (R^2^ = 60%, MPE = 90.8%); distance covered at high speed (R^2^ = 60.4%, MPE = 91.1%); and time spent at high speed (R^2^ = 60.2%, MPE = 91.2%). Finally, of the mixed metrics of TL, only session RPE interacted with the increase in hs-cTnT with a high likelihood (R^2^ = 65.8%, MPE = 96%), whereas Edwards’ TL (R^2^ = 58.1%, MPE = 71.6%) did not. 

## 4. Discussion

The aim of this study was to assess whether practical, field-based metrics of TL in soccer could be used to predict the exercise-induced release of hs-cTnT in healthy young players. Although previous studies have investigated exercise-induced hs-cTnT release in pubertal participants [12], studies involving team sports [28,29,30] and/or pre-pubertal participants are scarce [31]. Among these studies, only one, to our knowledge, reported hs-cTnT concentrations in subjects after playing soccer [30]. The novel finding of the present study is that common metrics of TL in soccer, easily obtained using player tracking systems, are strongly associated with the release of hs-cTnT in children and adolescents.

Our results coincide with previous research that observed an increase in hs-cTnT in apparently healthy young participants [12]. This increase was highly variable between participants (median increase = 5.96, IQR = 2.67–11.07) and supports previous observations of relatively high variability in hs-cTnT release at early ages [32]. However, our finding that older players (in both chronological and biological terms) are more likely to have higher hs-cTnT concentrations is not in line with the results of Tian et al. [33], who previously suggested that basketball players at Tanner stage II might reach higher peaks of hs-cTnT than those at Tanner stage III. This discrepancy might be related to a possible interaction between age and TL since more experienced players, from a technical and tactical perspective, might resolve real game situations more efficiently, requiring less intensity than a novice. However, the limited sample size in this study did not allow for multiple modeling to assess this potential interaction. In addition, hs-cTnT associations with age seem to differ depending on the age range observed. Studies that compared adolescents with adults found higher peaks in the younger participants [28,33], although this association remained inconclusive when comparing adolescents with children [33,34,35]. Fu et al. hypothesized that the higher variability in adolescents’ post-exercise hs-cTnT might be attributable to the immaturity of their myocardium since it would experience greater stress in response to an increased myocardial workload compared with that of adults [36].

Our main hypothesis was that hs-cTnT release is explained by iTL. Contrary to this hypothesis, only one HR metric (time spent in HR zone 5) explained the hs-cTnT increase, whereas the mean and peak HR, in both absolute and relative terms, did not. To our knowledge, this is the first study to include time spent in the maximal HR zone, which is a metric that provides information about not only the average and peak demands of an exercise (commonly reported as peak and mean HR) but also the distribution of intensities throughout a training session. The overall relevance of HR metrics during exercise in explaining the subsequent change in hs-cTnT is still under debate, especially in young participants. Previous research in adult populations found that hs-cTnT release after exercise was associated with the peak and average heart rate during exercise [11]. However, our results coincide with recent studies that did not find such an association when investigating the phenomenon in younger participants [33,37]. The acute cardiovascular response to exercise in children is more variable than that in adults; this variability is probably a consequence of anatomical, physiological, and psychological changes that occur during growth and maturation [38]. This high individual variation in exercise-induced HR might confound coaches when they estimate the actual TL from HR-based metrics. Consequently, other available metrics, such as RPE, speed, or distance, might be preferable when quantifying load in young players.

We also hypothesized that the external metrics of TL partially explain the variability of hs-cTnT over time. Our findings confirm that geospatial metrics during exercise are associated with the post-activity increase in hs-cTnT. These results concur with those of previous authors who associated the increase in cardiac biomarkers with indicators of exercise intensity, such as the time needed to cover a certain distance or the distance covered in a fixed time [11,36]. Confirming that hs-cTnT increases are proportional to the physical demands of an exercise would be compatible with the main hypothetical mechanisms for a reversible, physiological release of cTn induced by exercise [13,14]. On the other hand, to our knowledge, this is the first study to analyze the specific effects of exercise volume (time and distance) at maximal intensity. Accordingly, a novel finding of the present study is the likely positive association between the time or distance covered in the high-speed zone and the subsequent increase in hs-cTnT. This association implies that the hs-cTnT release depends not only on the overall intensity of the session but also on its distribution. Furthermore, the possible interaction between exercise duration and intensity was previously investigated in adolescents by Fu et al., who found that, although both parameters were associated with biomarker release, exercise intensity could better explain the increase in hs-cTnT [36].

In the present study, we assessed two mixed metrics of TL. When modeling the hs-cTnT variability using session RPE, we found a likely interaction with time (MPE = 96.6%), but, in contrast, the interaction between time and Edwards’ TL was uncertain (MPE = 72%). In both cases, these results coincide with the likelihood of the main variables in the calculations since RPE was shown to have a likely interaction with time, but heart rate indicators did not show an interaction. To the authors’ knowledge, this is the first study to report hs-cTnT associations with both eTL and iTL metrics in young soccer players. However, the association between external and internal TL was not addressed in this study and could be further explored in future research. In this regard, Akubat, Barrett, and Abt proposed the use of an external-internal TL ratio, which combines the individualized training impulse, high-intensity distance, and total distance [39]. This ratio has been previously associated with players’ total quality of recovery and muscle soreness, but its potential association with the release of myocardial markers is currently unknown [8].

From a clinical point of view, the strengths of the present study include the incorporation of practical, field-based TL metrics based on player tracking that are largely missing in clinical literature on hs-cTnT kinetics associated with exercise. In addition, from a practical perspective, our results raise the possibility of using post-exercise hs-cTnT elevations as a surrogate metric of TL. Both approaches might guide future research to evaluate the potential use of novel clinical markers, such as hs-cTnT, when quantifying players’ TL. However, several limitations of our study should be considered. First, we only studied a small sample of male soccer players at maturational stages from II to IV. Such a restriction prevents our results from being generalized to both sexes, other sports, or a wider range of maturational stages. For these reasons, future studies should address our limitations by encompassing larger and wider samples in terms of the number of participants, sex, maturational status, and sports modality. In addition, we only assessed a proportion of the available metrics for sports performance assessment. Therefore, future research might consider assessing other variables from both physiological (e.g., ventilatory thresholds or lactate accumulations) and analytical perspectives (e.g., accelerations and decelerations or number of impacts).

Our findings support the notion that children might present elevations of hs-cTnT related to exercise. This evidence could be considered in a clinical setting when interpreting hs-cTnT results in young athletes. In addition, we found that exercise duration or distance in high-intensity zones for both rHR and speed might be associated with the subsequent release of hs-cTnT. Since these measurements are commonly registered by smartwatches worn by athletes to monitor their sessions, our results might also be applicable to clinicians who consult the training history in their patients’ smartwatches, when available. Furthermore, the findings of the present study also apply in reverse since we demonstrated that sessions with a higher high-intensity density might be associated with a higher elevation of a cardiac-damage biomarker. Because the mechanisms of exercise-induced hs-cTnT are not completely understood, coaches who work with children should be aware that training sessions that require players to perform in high-intensity zones for long periods might induce the release of cardiac-damage markers such as hs-cTnT.

## 5. Conclusions

Our results confirm that healthy children have elevated blood concentrations of hs-cTnT after an intermittent small-sided soccer game. Players’ ages, maturational stages, and training experience were positively associated with the release of hs-cTnT. Internal TL metrics did not explain the elevations of the biomarker, with the exception of time spent in the maximal HR zone. The external metrics of TL were directly associated with the increase in hs-cTnT. A novel finding is that the time that young soccer players spend in the maximal heart rate or speed zones during a training session might be associated with an increase in hs-cTnT in the following 3 h.

## Figures and Tables

**Figure 1 ijerph-16-04853-f001:**
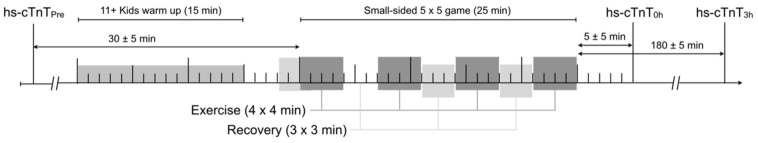
Timeline of procedures.

**Figure 2 ijerph-16-04853-f002:**
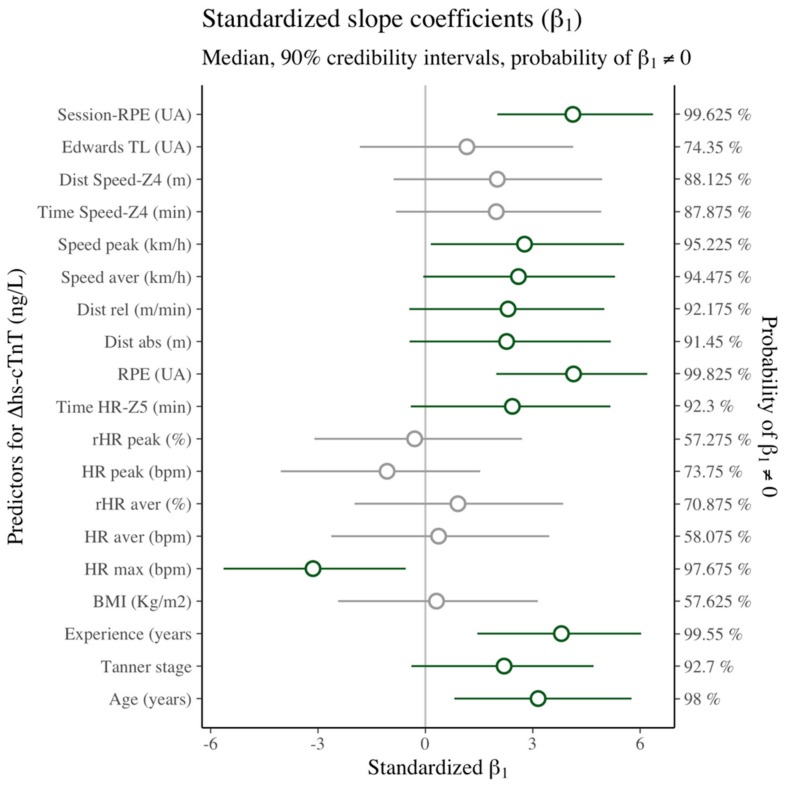
Standardized interaction coefficients (*β*_3_) by variable. Green intervals denote variables with Maximum Probability of Effect (MPE) values over 90%.

**Table 1 ijerph-16-04853-t001:** Participant characteristics.

Variable	Mean ± SD
Age (years)	11.9 ± 2
Tanner stage (*n*)	II = 8, III = 8, IV = 4
Height (cm)	151.2 ± 13.1
Weight (kg)	43.1 ± 13
BMI (kg/m^2^)	18.4 ± 2.62
Experience (years)	5.9 ± 1.7
HR max (bpm)	208 ± 2

Note: Tanner stage, was summarized using its frequency distribution. BMI, body mass index; HR max, maximum heart rate.

**Table 2 ijerph-16-04853-t002:** Summary of training load metrics.

Variable	Mean ± SD
Internal Training Load	
HR av (bpm)	181.17 ± 8.97
rHR av (% HR max)	87.15 ± 4.46
HR peak (bpm)	198.5 ± 7.32
rHR peak (% HR max)	95.48 ± 3.46
Time HR Zone 5 (min: s)	13:23 ± 1:15
∆HR 1 min recovery (bpm)	−49.86 ± 7.61
RPE (AU)	17.25 ± 0.91
External Training Load	
Distance (m)	2101.22 ± 217.31
rDistance (m min^−1^)	131.33 ± 13.58
Speed av (km h^−1^)	7.34 ± 0.79
Speed peak (km h^−1^)	22.15 ± 1.86
Time Speed Zone 4 (min)	6.48 ± 5.02
Distance Speed Zone 4 (min)	40.04 ± 31.42
Mixed Metrics of Training Load	
Session RPE (AU)	431.25 ± 22.76
Edwards’ TRIMP (AU)	6013 ± 381.64
Hs-cTnT	
Pre (ng/L)	1.5 (IQR = 1.5–1.5)
0 h (ng/L)	1.5 (IQR = 1.5–1.94)
3 h (ng/L)	7.46 (IQR = 4.17–12.57)

Note: High-sensitivity cardiac troponin T (hs-cTnT) is expressed as the mean [interquartile range]. HRav, average heart rate; rHRav, average relative heart rate; HRpeak, peak heart rate; rHRpeak, peak relative heart rate; Time HR Zone 5, time spent in heart rate zone 5; ∆HR 1 min recovery, heart rate decrease in one minute of recovery; RPE, rating of perceived exertion; rDistance, relative distance; Time Speed Zone 4, time spent in maximal speed zone; Distance Speed Zone 4, distance covered at maximal speed; TRIMP, Training impulse.

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
