# Peer review of "Effect of Training Load on Post-Exercise Cardiac Troponin T Elevations in Young Soccer Players"

_ijerph, 2019, doi:10.3390/ijerph16234853_

Round 1

Reviewer 1 Report

General comments

The article Effect of training load on post-exercise cardiac troponin T elevations in young soccer players aims to investigate the association between TL and post-exercise cTn in a group of male young soccer players. The article provides interesting issues concerning the relationship between measures of external and internal load, and a potential impact for professionals from different areas related to soccer performance. Some minor issues are pointed out below. Also, I suggest an English review by a native speaker.

Specific comments

Abstract

There is a need to increase the clarity of the information regarding the results and conclusion in the abstract. The reader is unable to easily understand the main contribution of the article.

Key-words are repeating the title. I recommend changing it.

Introduction

In general, it is clear and provides us with the most relevant insights to support the research problem. Some minor issues are presented below.

Lines 45-47: what are the criteria for defining this diversity? Since you are talking about specific constructs of specific instruments, you should refer to the literature to support your assumptions.

I recommend you to clearly point out what is the advantage of the assessment of cTn in comparison to the currently available measures. If this advance is not clear, you are unable to justify the current study.

Experimental section

Some questions must be addressed by the authors in order to make the procedures clearer and increase the study's reproducibility.

How the sample was estimated?

A control placebo group is not necessary at the current design?

What is the validity and reliability of the GPS device used? This is mandatory information.

What is the sample of data acquisition of the GPS? Does it come with an accelerometer?

Are the players familiar with the RPE data collection?

Table 1 presents data regarding participants’ maximum heart rate. How this information was obtained?

The variables “relative distance covered per minute of exercise”, absolute distance” and “average speed during exercise” are supposed to give the same information since all SSG has the same duration. So, I recommend choosing just one. Besides, when analyzing the results (Appendix 1), the MPE is almost the same to them, which confirms this rationale.

For this specific public (U-11/U-12 players mostly, based on the sample characteristic), 18 km/h seems not achievable. What was the criterion to select this as the threshold for defining the high-intensity zone?

Some details about the small-sided games are missing. For example:

                - Was the offside rule adopted?

                - How teams were composed?

                - Teams were kept the same during all bouts?

What is the criterion to select a 30x20 field for a 5vs5 format? I consider this a too short pitch.

Results

Notes are missing in table 2.

There is an error in the title of figure 2.

Discussion

Lines 272-273: “perspective, our results raise the possibility of using post-exercise hs-cTnT elevations as a surrogate metric of TL”. Why replacing the usual measures of TL? Sometimes is more invasive (and time-spending) collecting blood samples than simply measuring the TL by GPS-based measures. This suggestion does not make any sense to me.

Lines 283-284: “Our findings confirm that children attending emergency department with chest pain after  training might present elevations of hs-cTnT related with the exercise” You didn’t test this, so this is only a speculation, which must be avoided.

References

19 from 39 (about 48%) last 5 years. Maybe the authors could search for more up-to-date references.

Some references don’t follow the Journal’s guidelines.

There are no references from IJERPH. Is this topic really covered by the Journal’s scope?

Author Response

We thang the reviewer for his/her effort reviewing our manuscript. All suggestions were appropriate and helped us to improve the quality of the article. Also, as the reviewer suggested, after including our modifications the article was submitted to the journal language editing service. Following you will find a detailed list of our modifications in the main text, that was updated. These modifications were highlighted in yellow in the main text as well.

COMMENT 1: There is a need to increase the clarity of the information regarding the results and conclusion in the abstract. The reader is unable to easily understand the main contribution of the article.

RESPONSE 1: We thank the reviewer for this suggestion, and definitively agree that it was unclear. In the new version of the manuscript, lines 23-27 were rewritten.

COMMENT 2: Key-words are repeating the title. I recommend changing it.

RESPONSE 2: We thank the reviewer for detecting these duplicate terms. Keywords were updated to cardiac biomarkers, exercise physiology and maturation.

COMMENT 3: Lines 45-47: what are the criteria for defining this diversity? Since you are talking about specific constructs of specific instruments, you should refer to the literature to support your assumptions.

RESPONSE 3: We agree with this suggestion. Citations were provided in line 54.

COMMENT 4: I recommend you to clearly point out what is the advantage of the assessment of cTn in comparison to the currently available measures. If this advance is not clear, you are unable to justify the current study.

RESPONSE 4: We are thankful for this comment, and reworded the last paragraph of the introduction to highlight the interest of this study.

COMMENT 5: How the sample was estimated?

COMMENT 5: Sample size was not stablished a priori, but were all includible children and parents disposed to participate in the study. We stated that our sample size was defined by convenience at the beginning of participants subsection.

COMMENT 6: A control placebo group is not necessary at the current design?

COMMENT 6: Since previous research was consistent reporting that a) hs-cTnT resting values are constant and normally undetectable, and b) exercise induces elevations in most of the healthy athletes, we opted for using the entire available sample as a single exercising group.

COMMENT 7: What is the validity and reliability of the GPS device used? This is mandatory information.

RESPONSE 7: We understand the relevance of reporting devices validity. Results of both validation articles we cited were included into the text.

COMMENT 8: What is the sample of data acquisition of the GPS? Does it come with an accelerometer?

RESPONSE 8: WIMU has both, a GPS (10 Hz) and an accelerometer (up to 1000 Hz). Since we used only GPS data, this sampling rate was included in line 112.

COMMENT 9: Are the players familiar with the RPE data collection?

RESPONSE 9: We thank the reviewer for detecting this missing statement. All participants in the campus were familiarized with RPE reporting. We added this statement within the text.

COMMENT 10: Table 1 presents data regarding participants’ maximum heart rate. How this information was obtained?

RESPONSE 10: Heart rate was calculated using Tanaka’s formula of 208 – (0.7 x Age). In the revised version of the manuscript we reported this in line 134.

COMMENT 11: The variables “relative distance covered per minute of exercise”, absolute distance” and “average speed during exercise” are supposed to give the same information since all SSG has the same duration. So, I recommend choosing just one. Besides, when analyzing the results (Appendix 1), the MPE is almost the same to them, which confirms this rationale.

RESPONSE 11: We understand the rationale of removing those variables and thank the reviewer for this suggestion. Although they are indeed related and the association might be redundant, differentiating such metrics might facilitate a reader’s interpretation of the same phenomenon in terms of distance or speeds. For this reason, we opted for preserving all variables.

COMMENT 12: For this specific public (U-11/U-12 players mostly, based on the sample characteristic), 18 km/h seems not achievable. What was the criterion to select this as the threshold for defining the high-intensity zone?

RESPONSE 12: This decision was based on the study of Whitehead, Till, Weaving, and Jones (2018) [15].

COMMENT 13: Some details about the small-sided games are missing. For example: Was the offside rule adopted? How teams were composed? Teams were kept the same during all bouts?

RESPONSE 13: We thank the reviewer for all three suggestions. This understand the relevance of this information and included all in lines 199-212.

COMMENT 14: What is the criterion to select a 30x20 field for a 5vs5 format? I consider this a too short pitch.

RESPONSE 14: This decision was made based on Mascarin et al (2018) [25].

COMMENT 15: Notes are missing in table 2.

RESPONSE 15: We thank the reviewer for pointing this out. In the new version of the manuscript tables have a proper legend.

COMMENT 16: There is an error in the title of figure 2.

RESPONSE 16: The manuscript was sent for language editing.

COMMENT 17: Lines 272-273: “perspective, our results raise the possibility of using post-exercise hs-cTnT elevations as a surrogate metric of TL”. Why replacing the usual measures of TL? Sometimes is more invasive (and time-spending) collecting blood samples than simply measuring the TL by GPS-based measures. This suggestion does not make any sense to me.

RESPONSE 17: We agree with this reviewer concern. For this reason, we avoided any suggestion of method replacement in our manuscript.

COMMENT 18: Lines 283-284: “Our findings confirm that children attending emergency department with chest pain after training might present elevations of hs-cTnT related with the exercise” You didn’t test this, so this is only a speculation, which must be avoided.

RESPONSE 18: We thank the reviewer for this suggestion. The sentence in line 395 was updated.

COMMENT 19: 19 from 39 (about 48%) last 5 years. Maybe the authors could search for more up-to-date references.

RESPONSE 19: We thank the reviewer for this suggestion. We revised our references and agree to preserve them since although some of them are not recent, all cases are pertinent to the content where they were cited.

COMMENT 20: Some references don’t follow the Journal’s guidelines.

RESPONSE 20: All references are now revised. Changes are highlighted in yellow.

COMMENT 21: There are no references from IJERPH. Is this topic really covered by the Journal’s scope?

RESPONSE 21: This manuscript was submitted for a special issue entitled: Health, Exercise and Sports Performance.

Reviewer 2 Report

In this study, authors evaluated how TL measurements during a small sided soccer game were related to the subsequent increase cTn in young players. The results are interesting and would be helpful to readers, if some concerns below can be addressed.

In abstract section, please check the unit of MPE (line 24, line 25). Reference citation should be before the period because the citation is considered part of the sentence. Please check the superscript of unit (line 122-124). Please check the symbols in Fig. 2. Remove the range of mean±SD in table 1, table 2 and in line 155. In line 170, what’s the LoD ? Please revise the (90% CI 52.5% to 74.7%) in line 176 to (90% CI= 52.5% - 74.7%).

Author Response

We thang the reviewer for his/her effort reviewing our manuscript. All suggestions were appropriate and helped us to improve the quality of the article. Also, as the reviewer suggested, after including our modifications the article was submitted to the journal language editing service. Following you will find a detailed list of our modifications in the text. These modifications were highlighted in yellow in the main text as well.

COMMENT 1: In abstract section, please check the unit of MPE (line 24, line 25).

RESPONSE 1: We thank the reviewer for spotting these missing units. Now all MPE are correctly reported.

COMMENT 2: Reference citation should be before the period because the citation is considered part of the sentence.

RESPONSE 2: All citations were moved inside the sentences.

COMMENT 3: Please check the superscript of unit (line 122-124).

RESPONSE 3: Superscripts were revised.

COMMENT 4: Please check the symbols in Fig. 2.

RESPONSE 4: The beta coefficient we are visualizing is indeed number 3, namely the interaction, as described in the statistical analysis subsection.

COMMENT 5: Remove the range of mean±SD in table 1, table 2 and in line 155.

RESPONSE 5: We accept the reviewer suggestion, and ranges have been removed.

COMMENT 6: In line 170, what’s the LoD ?

RESPONSE 6: We thank the reviewer for detecting missing statements before abbreviations. Limit of Detection (LoD) and Upper Reference Limit (URL) were introduced in the text.

COMMENT 7: Please revise the (90% CI 52.5% to 74.7%) in line 176 to (90% CI= 52.5% - 74.7%).

RESPONSE 7: These confidence intervals were revised

Reviewer 3 Report

The introduction is overly extensive and not narrowly focused.  All table and figures need legends that define abbreviations. This was a well-performed study, and the findings have interesting implications. However, the underlying premise in the introduction and the application in the discussion do not seem to agree. Please state more clearly the reason for the current study, the novelty of the current study, and the applicability of the current study. For instance, what is the usefulness of measuring troponin in athletes following training?

There are multiple English language errors or awkward expressions throughout the manuscript. I recommend that this be reviewed by a professional editing service. For example, in just the first paragraph of the introduction, I found the following problems:

The first sentence is awkwardly worded. The third sentence is unclear due to inappropriate comma usage and an unclear subject. The sentence starting at the end of line 41 has a comma between the subject and verb ("These measures, might be denominated").

These and other errors must be fixed before publication.

Author Response

We thank the reviewer for his/her effort reviewing our manuscript. All suggestions were appropriate and helped us to improve the quality of the article. Also, as the reviewer suggested, after including our modifications the article was submitted to the journal language editing service. Following you will find a detailed list of our modifications in the text. These modifications were highlighted in yellow in the main text as well.

COMMENT 1: The introduction is overly extensive and not narrowly focused. This was a well-performed study, and the findings have interesting implications. However, the underlying premise in the introduction and the application in the discussion do not seem to agree. Please state more clearly the reason for the current study, the novelty of the current study, and the applicability of the current study. For instance, what is the usefulness of measuring troponin in athletes following training? 

RESPONSE 1: We thank the reviewer for this comment. The introduction was shortened and we made it more specific. In addition, the full text was also sent to the journal for language editing.

COMMENT 2: All table and figures need legends that define abbreviations.

RESPONSE 2: We agree with the reviewer suggestion. In the updated version of the manuscript tables and figures are properly noted.

COMMENT 3: There are multiple English language errors or awkward expressions throughout the manuscript. I recommend that this be reviewed by a professional editing service. For example, in just the first paragraph of the introduction, I found the following problems: The first sentence is awkwardly worded. The third sentence is unclear due to inappropriate comma usage and an unclear subject. The sentence starting at the end of line 41 has a comma between the subject and verb ("These measures, might be denominated"). These and other errors must be fixed before publication.

RESPONSE 3: We agree with the reviewer. The new version of the manuscript was submitted to the journal language editing service.